# Immunological Targets of Biologic Drugs in Allergic Skin Diseases in Children

**DOI:** 10.3390/biomedicines9111615

**Published:** 2021-11-04

**Authors:** Paola Di Filippo, Daniele Russo, Marina Attanasi, Sabrina Di Pillo, Francesco Chiarelli

**Affiliations:** 1Department of Pediatrics, University of Chieti, 66100 Chieti, Italy; danielerusso1607@gmail.com (D.R.); marina_attanasi@hotmail.it (M.A.); sabrinadipillo@gmail.com (S.D.P.); chiarelli@unich.it (F.C.); 2Center of Excellence on Aging, University of Chieti, 66100 Chieti, Italy

**Keywords:** atopic dermatitis, chronic urticaria, children, biologics, molecular pathways, IgE

## Abstract

Atopic dermatitis and urticaria are two invalidating skin disorders that are very common in children. Recent advances in the understanding of their specific intracellular molecular pathways have permitted the development of precise biological molecules, targeting inflammatory mediators and arresting the pathogenetic pathways of skin diseases. Many biologics with promising results have been studied, although few are currently approved in children. In this review, we aim to provide the latest evidence about the use, indications, efficacy and safety of biologic therapies to treat atopic dermatitis and chronic urticaria in children and adolescents.

## 1. Introduction

Currently, the most common treatment strategies for atopic dermatitis (AD) and urticaria in children focus on relieving symptoms and reducing inflammation rather than treating the underlying cause. During the last 10 years, therapy options for pediatric skin diseases were improved considerably, especially thanks to new biologic drugs targeting IgE, IL-4, IL-5, and IL-13.

Monoclonal antibodies (mAbs) are monovalent antibodies that bind to the same epitope with an exquisite targeted selectivity and, therefore, a lower toxicity. mAbs are an innovative therapy for several diseases and their applications are constantly extending, including skin diseases such as AD and urticaria [1].

This review focuses on the new insights about biologic drugs for AD and urticaria in children, whose mechanism of action is illustrated in Figure 1. 

## 2. Atopic Dermatitis

Atopic dermatitis, also known as eczema and atopic eczema, is a chronic inflammatory skin disorder [2] and affects up to 20% of children and 10% of adults in high-income countries [3,4]. Although the prevalence of AD has reached a stable level in many high-income countries, it is increasing in low-income and middle-income countries, probably because of many environmental factors [5]. The increasing prevalence is probably due to the so-called hygiene hypothesis, supported by an inverse socioeconomic gradient and an association with numbers of siblings [6,7,8].

AD can occur at any age with a peak in early childhood (typically at age 3–6 months), but is common in adults as well, including both persistent and new-onset disorders [9,10,11,12]. Predictors of persistent AD into adulthood include concurrent asthma, hay fever, young age at onset, low socioeconomic status, and non-white ethnicity. To date, it is unclear if these predictors are independent of disease severity [9,13]. As a matter of fact, the causes and outcomes of adult AD need to be better understood

The old definition of AD as a disease that resolves in early childhood was replaced by its more recent definition as disorder that can differently evolve, ranging from early transient disease to relapsing-remitting AD, chronic persistent AD, or long periods of remission followed by recurrence [14,15].

The pathogenesis of AD is multifactorial, including both genetic and environmental factors. Loss-of-function mutations in the gene encoding filaggrin, a major structural protein in the epidermis, are the most frequently reported genetic variants, supporting its key role in the skin barrier function [16]. A complex interaction between a dysfunctional skin barrier, skin microbiome abnormalities, and a predominantly type-2-skewed immune dysregulation plays an essential role in the establishment of the disease [17]. These mechanistic drivers can promote and interact with others. For example, skin barrier weakness that is attributable to a filaggrin deficiency promotes inflammation and T-cell infiltration; colonization or infection with Staphylococcus aureus damages the skin barrier and induces inflammatory responses; local Th2 immune responses further reduce skin barrier function, drive itching, and facilitate dysbiosis in favor of members of the genus Staphylococcus, particularly S aureus [2]. AD is characterized by intense itching and recurrent eczematous lesions, although its clinical presentation may be heterogeneous [6]. Despite the mast cells being classically defined as the main type of cells responsible for itching, recent studies reassessed the role of basophils [18,19]. Basophils could induce pruritus through their expression of a multitude of pruritogens, such as IL-31, histamine, and Th2 cytokines. [18] Since patients with allergen-specific IgE are more likely to experience itching flare-ups than those without allergen-specific IgE, it was recently hypothesized that allergen exposure drives acute itching flare-ups. A recent study generated a murine model of AD-like disease in which challenge with a model allergen elicits acute itching flare-ups. [19] Although dependent on IgE, the authors found that itching flare-ups occurred independently of tissue-resident mast cells, but they were critically dependent on basophils activation in mice. Therefore, both mast cells and basophils can induce itching in response to the same allergen, but the setting of AD-associated inflammation leads to basophils upregulating the FcεRIa of IgE receptors, enhancing their capacity to mediate atopic itching flare-ups. [19] Since AD is a T-cell driven disease characterized by a strong activation of Th2 immune response and its cytokines, targeting type-2 therapy seems to be a rational strategy. AD can be extrinsic or intrinsic based on the presence of increased total and allergen-specific IgE levels, higher rates of eosinophils, and a family history of atopic disease [20]. IgE levels are increased in 80% of patients with extrinsic AD, and IgEs were initially proposed as a valid therapeutic target, but their pathogenic role in AD remains unclear [21]. Moreover, eosinophils, whose blood and skin levels can be increased in AD patients, may not play a critical role in the pathogenesis of AD, as demonstrated by the ineffectiveness of mepolizumab in AD. [20] Although both intrinsic and extrinsic subtypes are characterized by a strong Th2 activation, there is increasing evidence that the pathophysiology involves multiple immune pathways and that the Th22, Th17/IL-23, and Th1 cytokine pathways also have a pivotal role in some AD subtypes. [20] For example, intrinsic AD shows a stronger activation of Th17 and Th22 responses compared to extrinsic AD. [20] In acute lesions, the activation of Th2 (IL-4, IL-5, IL-13, IL-31, and CCL18) and Th22 (IL-22 and S100A proteins) pathways down-regulates terminal differentiation genes and tight junction products, contributing to the skin barrier defect in AD. In chronic AD, Th2 and Th22 responses are intensified but there is also a simultaneous activation of the Th1 axis (IFN-γ, CXCL9, and CXCL10) [20].

Therefore, new therapies targeting specific inflammatory pathways (such as Th1, Th17, and Th22) through modulation of cytokines, receptors, and other molecules, represent a promising strategy for the individualized treatment of AD [22]. 

## 3. Biologics in Atopic Dermatitis

### 3.1. Omalizumab

Omalizumab is an anti-IgE mAb that binds the FcεRI receptor, blocking IgE, reducing circulating IgE levels, and inhibiting basophils and mast cells [23]. Several studies have shown the ineffectiveness of omalizumab as a therapeutic agent in AD [24,25], suggesting that increased IgE levels could be an epiphenomenon of AD, mediating comorbidities such as food allergies, asthma, and rhinoconjunctivitis, but not the AD itself [20].

Furthermore, omalizumab therapy in AD was related to several problems. First, IgE is not the only pathogenetic mediator in AD; therefore, anti-IgE therapy may not be effective [26]. Secondly, most AD patients have higher IgE levels than the recommended limit for asthma treatment (700 IU/mL), but higher doses could be associated with side effects, such as a higher risk of anaphylaxis [27]. Ultimately, omalizumab is a very costly therapy and the cost/benefit ratio must be considered [28]. Therefore, omalizumab is not currently recommended for AD treatment, given its inefficacy in reducing chronic skin inflammation in AD [22]. Considering the emerging role of basophils in atopic itching flare-ups associated with allergen exposure [18], it would be desirable for anti-IgE therapy to be specifically assessed in clinical trials as a therapeutic agent for AD-associated itching.

### 3.2. Dupilumab

Among the Th2 immune mediators, IL-4 and IL-13 play a key role in the pathogenesis of AD and their genetic polymorphisms are associated with AD. IL-4 and IL-13 decrease the expression of genes that encode for essential components of the epidermal barrier (such as filaggrin, loricrin, and involucrin), compromising the skin barrier function and promoting the penetration of bacteria and allergens into the skin, leading to infections and allergen sensitization [20]. Furthermore, IL-4 and IL-13 inhibit the skin production of antimicrobial peptides and, thus, predispose AD skin to colonization and infection of Staphylococcus aureus, which further worsens skin inflammation and barrier defects [20].

The biological function of IL-4 and IL-13 is expressed through the binding of two receptor subtypes (IL4R): the type I receptor binds only to IL-4 and is composed by the heterodimer IL-4Rα/gc, while the type II receptor binds to both IL-4 and IL-13 and is composed of IL-4Rα/IL-13Rα1 [29]. Dupilumab is a fully humanized monoclonal antibody acting against the alpha subunit of the IL-4 receptor (IL4Rα), blocking IL-4 and IL-13 receptors and signaling [30]. 

Many phase 2 and 3 clinical trials demonstrated its effectiveness in improving the skin symptoms and the quality of life in AD [31,32,33,34,35]. Therefore, dupilumab was approved by the Food and Drug Administration (FDA) in March 2017 and by European Medicines Agency (EMA) in September 2017, being the first targeted biologic therapy for adults with moderate-to-severe AD [22].

LIBERTY AD CHRONOS is a 1-year, randomized, double-blinded, placebo-controlled, phase 3 trial, which demonstrated the long-term efficacy and safety of dupilumab with topical corticosteroids (TCS) versus placebo with TCS in 740 adults with moderate-to-severe AD [36]. 

Furthermore, the data show that the response was maintained for at least 1 year of continuous treatment [21]. The long-term use of dupilumab was evaluated and its sustained efficacy over a 76-weeks treatment period was demonstrated in 1491 adults with AD. Furthermore, further therapy for AD during the treatment period was not required in 50.3% of patients, demonstrating that dupilumab monotherapy, alone or in association with topical AD medications, provides long-term disease control [37].

Moreover, dupilumab demonstrated a good safety profile. Most of the observed side-effects were mild and not dose-limiting. The most common side-effects were injection-site reactions, conjunctivitis, and upper respiratory tract infections [36], and they were reported more often at the beginning of treatment and diminished over time [37].

Recent insights also suggested the role of dupilumab in modulating the skin microbiome. In AD, Th2 cytokines induce skin barrier function alterations, facilitating Staphylococcus aureus colonization and a lower skin microbial diversity [38]. The colonization by Staphylococcus aureus worsens the inflammatory state and disease severity [39]. AD-1307 EXPLORE trial demonstrated the role of dupilumab in the normalization of skin barrier function, inhibiting type 2 cytokines and inducing a progressive shift from a lesional to a nonlesional molecular phenotype [40]. Additional findings regarding the role of dupilumab in skin microbiome were successively reported in AD-LIBERTY EXPLORE trial [39]. This trial confirmed that treatment with subcutaneous dupilumab for 16 weeks in 27 adult patients increases microbial diversity and reduces Staphylococcus aureus colonization compared to 27 placebo-treated patients [39].

Dupilumab was successively approved in teenagers between 12 and 18 years old and it was recently approved by the EMA in children from 6 years of age with moderate-to-severe AD when topical therapies are insufficient or not recommended [22]. It was recommended a subcutaneous administration with a 400-mg loading dose followed by 200 mg every 2 weeks in teenagers with body weights of less than 60 kg, and a 600-mg loading dose followed by 300 mg every 2 weeks was recommended in teenagers weighing 60 kg or more [41]. In the LIBERTY AD ADOL randomized phase III clinical trial, the greater efficacy of the every-2-week regimen compared to the every-4-week regimen was recently demonstrated. Furthermore, pharmacokinetic data supported that the every-2-week regimen provided higher dupilumab trough concentrations [41]. 

Studies evaluating efficacy and safety in pediatric age are lacking, especially for subjects under 12 years old. However, new information was recently gained from three randomized, double blinded, placebo-controlled, phase II and III trials.

In a multicenter, phase IIa, open-label study involving 38 children aged between 6 and 12 years, a single-dose of dupilumab promptly improved AD with further improvements through week 52. Side effects were mild-to-moderate and transient, and a treatment discontinuation was not necessary [42], confirming a similar safety profile as in adults. 

Dupilumab proved to be effective and well tolerated in LIBERTY AD PEDS [43], a recent double-blind, 16-week, phase III trial, involving 367 children with severe AD aged between 6 and 11 years. The patients were randomized 1:1:1 to 300 mg dupilumab every 4 weeks (300 mg q4w), a weight-based regimen of dupilumab every 2 weeks (100 mg q2w if baseline weight < 30 kg; 200 mg q2w if baseline weight ≥ 30 kg), or placebo; dupilumab was administered in association with medium-potency TCS. Both the q4w and q2w dupilumab + TCS regimens resulted in a statistically significant improvement in signs, symptoms and quality of life compared to the placebo + TCS group. Total or near total clearance of skin lesions was demonstrated in 33% and 30% of patients treated with dupilumab every 4 weeks and every 2 weeks, respectively, compared to 11% of patients of the placebo group (*p* < 0.0001 and *p* = 0.0004, respectively). In addition, dupilumab showed a significantly improved quality of life, reducing patient’s itching and anxiety or depression of the patients and their parents. The main side effects reported were conjunctivitis, keratoconjunctivitis and injection site reactions. Response to therapy was weight-dependent: optimal doses were 300 mg every 4 weeks in children < 30 kg and 200 mg every 2 weeks in children ≥ 30 kg [43].

LIBERTY AD PRESCHOOL is an open-label, multicenter, phase II study that included a cohort of children aged between 2 and 6 years old and a second cohort of children aged between 6 months and 2 years of age. A single dose of dupilumab reduced signs and symptoms of AD and was well tolerated. Furthermore, a slightly better response was seen in older children compared to younger ones [44].

All these safety and efficacy results support the use of dupilumab as a long-term treatment for children with severe AD and led to its approval in 2020 by the FDA and the EMA in patients from 6 years of age with moderate-to-severe AD when topical therapy is insufficient or not recommended. [45]

### 3.3. Mepolizumab

Mepolizumab is a fully humanized monoclonal anti-IL-5 antibody. It acts against hypereosinophilia, and thus, it was approved for severe eosinophilic asthma [46]. Considering that AD is characterized by the expression of Th2 cytokines, including IL-5 and eosinophil infiltration [46], several trials were performed on mepolizumab, but the results are still unclear. 

A prompt reduction in peripheral blood eosinophils was observed after two rounds of administration of mepolizumab in patients with severe AD, but the clinical results were unsatisfactory [46] and no effect on atopy patch test reactions was observed [47], suggesting that increased eosinophils levels could be an epiphenomenon of AD [20]. Long-term trials, preferably stratifying patients based on eosinophils level, are needed to clarify its role in AD treatment.

### 3.4. Tezepelumab and Etokimab

The epithelial cell-derived cytokines IL-33 and thymic stromal lymphopoietin (TSLP) act upstream of effector cytokines (such as IL-4, IL-13, and IL-31); therefore, they could be excellent targets in AD [23]. TSLP is crucial in the upregulation of IL-13, IgE, and chemokine (C-C motif) ligand 17/thymus, as well as activation-regulated chemokines (CCL17/TARC) [48]. 

TSLP serum values in AD patients are higher compared to healthy controls; thus, it was proposed as a target to control inflammation in AD [49]. Tezepelumab is a humanized monoclonal antibody that binds TSLP and prevents its interaction with the receptor complex. In a phase II RCT, 111 patients with moderate-to-severe AD, treated with topical steroids, received either 280 mg tezepelumab subcutaneously every 2 weeks or a placebo. After 12—and especially after 16—weeks of therapy, a reduction in the Eczema Area and Severity Index was demonstrated, but the improvement was not statistically significant compared to placebo [49]. Further studies are needed to establish its efficacy in the AD treatment.

Etokimab is a monoclonal antibody that acts against IL-33. In a phase II study, patients who received a single dose of etokimab showed a significant improvement in their EASI scores, but a placebo group was not established [50]. Tezepelumab and etokimab are exciting therapeutic agents, but relevant data are still lacking, and further studies are needed to validate their efficacy and safety.

### 3.5. Nemolizumab

In patients with AD, increased IL-31 levels were found. IL-31 plays an important role in mediating the pruritus [20,51] that stimulates the exacerbation of AD and sleeping disorders, with a negative impact on the patients’ quality of life [52]. 

Nemolizumab is a humanized monoclonal antibody that acts against the IL-31 receptor. Recently, a significant clinical improvement, especially of pruritus, was demonstrated in adult patients with moderate-to-severe AD. Specifically, the pruritus visual-analogue scale score improved from baseline in 63.1% of patients treated with nemolizumab compared to 20.9% of patients treated with placebo in a randomized-controlled 12-week trial [53]. 

Recently, in a 24-week, randomized, double-blind, multicenter study involving 226 adults with moderate-to-severe AD, nemolizumab administration at a dosage of 10, 30, and 90 mg was compared to placebo. Nemolizumab administration resulted in rapid and sustained improvement of cutaneous manifestations and pruritus, and the maximal efficacy was observed at 30 mg. Furthermore, a good safety profile of nemolizumab was shown; the most common side effects observed were nasopharyngitis and upper respiratory tract infections [54].

Currently, nemolizumab is still not approved for any indication. Longer trials, also involving children, are required to assess its long-term effect and its safety in children.

### 3.6. Lebrikizumab and Tralokinumab

IL-13 plays a pivotal role in the Th-2 immune response. Lebrikizumab is a monoclonal antibody that binds soluble IL-13, preventing the heterodimerization of IL-13Ra1/IL-4Ra and the signaling that follows [55]. 

TREBLE is a phase II RCT performed in adult patients with moderate-to-severe treatment-unresponsive AD; treatment with 125 mg of Lebrikizumab every 4 weeks was associated with early symptom improvement and an acceptable safety and tolerability profile [55]. Recently, a phase IIb RCT involving 280 adult patients with moderate-to-severe AD confirmed that lebrikizumab provides a rapid and dose-dependent improvement in AD clinical manifestations during 16 weeks of treatment, with a favorable safety profile [56]. Tralokinumab is a humanized IL-13-neutralizing monoclonal antibody. A phase IIb RCT demonstrated its efficacy and safety in moderate–severe AD adult patients at a dosage of 300 mg every two weeks in association with TCS [57]. Its combined use with TCS compromises the efficacy assessment of tralokinumab alone. More recently, two phase III RCTs (ECZTRA 1 and ECZTRA 2) including adults with moderate-to-severe AD, randomized to subcutaneous tralokinumab 300 mg every 2 weeks or placebo, showed that tralokinumab monotherapy was superior compared to placebo at 16 weeks of treatment and was well tolerated up to 52 weeks of treatment [58].

### 3.7. OX40 Inhibitors

Keratinocytes and Langerhans cells in the lesional skin of AD patients highly express TSLP, triggering the expression of OX40L on dendritic cells. Therefore, the TSLP-OX40 ligand (OX40L) pathway seems to be an initiation factor for Th2 immune activation. OX40 is a costimulatory receptor expressed on activated T cells and the OX40–OX40L interaction is important in the generation and maintenance of Th2 responses in several allergic conditions such as allergic asthma, rhinitis, and conjunctivitis [20].

GBR 830 is a humanized monoclonal antibody that acts against OX40. A phase II study found that two intravenous administrations every 4 weeks were well tolerated and induced a significant clinical improvement, a reduction in the epidermal thickness, and progressive tissue changes, which were highlighted in the biopsy specimens by a reduction in Th1 (IFN-γ/CXCL10), Th2 (IL-31/CCL11/CCL17), and Th17/Th22 (IL-23p19/IL-8/S100A12) mRNA expression in lesional skin [59]. This is the only trial that evaluated the efficacy of GBR 830 in AD. Therefore, more studies are needed to better define its role in the management of AD.

### 3.8. Fezakinumab 

Fezakinumab selectively inhibits IL-22, a cytokine involved in skin barrier dysfunction and epidermal hyperplasia. [20] In a phase II study performed in patients with moderate–severe AD stratified according to their skin IL-22 levels, fezakinumab significantly reduced the SCORAD index. Fezakinumab also demonstrated a good safety profile: upper respiratory tract infections were shown to be the most common adverse effects [60]. A randomized, placebo-controlled, multicenter, phase IIa clinical trial involving 59 patients with moderate-to-severe AD assessed lesional and nonlesional skin biopsy specimens obtained before (baseline), during (week 4), and after (week 12) the treatment with fezakinumab versus placebo (2:1). Fezakinumab treatment resulted in the suppression of the mRNA expression of multiple genes related to the Th1, Th2, Th17, and Th22 pathways. Furthermore, considering the efficacy of IL-22 inhibition only in patients with severe AD, patients were stratified according to the baseline IL-22 mRNA expression. A greater mean transcriptomic improvement was found in the IL-22-high drug-treated group (82.8% and 139.4% at 4 and 12 weeks, respectively) compared to the IL-22-high placebo-treated group (39.6% and 56.3% at 4 and 12 weeks, respectively) or the IL-22-low groups [61].

### 3.9. JAK Inhibitors

In AD, activation of the JAK-STAT signaling pathway induces the polarization of Th2 and the disruption of the skin barrier, activates eosinophils and B cell maturation, increases epidermal chemokines, and reduces AMPs [20].

JAK inhibitors interfere with the JAK enzyme and, thus, the JAK/STAT signaling pathway, inhibiting the activity of several cytokines and growth factors involved in inflammatory and cell replication processes [62].

First generation JAK inhibitors (such as baricitinib) target more than one JAK enzyme, while second or newer generation JAK inhibitors (such as upadacitinib and abrocitinib) target specific JAK enzymes, minimizing the effects related to the inhibition of JAK2 and JAK3 [63].

Several JAK inhibitors are currently under study for the treatment of AD.

Baricitinib inhibits JAK1 and JAK2 enzymes and was the first JAK-inhibitor studied for AD [63]. A phase III study demonstrated the efficacy of baricitinib in the treatment of moderate–severe AD in adults [64], but it seems to reduce fertility and induce a teratogenic effect, albeit at a dosage 20 times higher than that recommended for the treatment of AD [65]. 

Upadacitinib and abrocitinib are two second generation JAK-inhibitors that selectively inhibit the JAK1 enzyme [66]. A phase IIb study found a significant clinical improvement defined by a significant EASI score reduction in patients treated with upadacitinib compared to placebo, demonstrating its efficacy in patients with moderate–severe AD [67]. Nevertheless, upadacitinib seems to induce teratogenic effects on animals, and thus, the administration to fertile women should approached with caution [68].

A phase IIb study found a significant EASI score reduction in adults with moderate-to-severe AD receiving abrocitinib 200 mg compared to placebo [68].

## 4. Conclusions Regarding the Use of Biologic Drugs for Atopic Dermatitis in Children

Currently, dupilumab is the only biologic drug with strong evidence of efficacy in AD, which reflects the key role of IL-4 and IL-13 in the pathogenesis of AD. At present, dupilumab is approved for children >6 years of age. A sufficient efficacy in AD was not demonstrated for omalizumab and mepolizumab, reflecting the possible elusive role of IgE and eosinophils in the pathogenesis of clinical manifestations of AD. Nemolizumab, lebrikizumab, etokimab, fezakinumab, and tralokinumab seem to be promising biologic drugs against AD, but longer follow-up and larger studies assessing their efficacy and safety profile are needed.

Moreover, it is important to recognize that AD is a heterogeneous disease and the identification of patient subgroups based on immunological characteristics would allow a tailored treatment. For example, the stronger activation of Th17 and Th22 responses in intrinsic compared to extrinsic AD could predict a better response of fezakinumab in this subgroup of patients. Indeed, several biologic drugs demonstrated efficacy and safety in clinical trials that included AD patients, but a significant proportion of patients were poor responders. A recent model-based meta-analysis of clinical trials allowed the development of a mathematical model that reproduced the reported clinical efficacy of nine biological drugs (dupilumab, lebrikizumab, tralokinumab, secukinumab, fezakinumab, nemolizumab, tezepelumab, GBR 830, and recombinant interferon-gamma) by describing the system-level pathogenesis of AD. Dupilumab and lebrikizumab showed the highest efficacy, suggesting that the IL-13 has the highest contribution in the pathogenesis of AD among the evaluated drug targets, and that baseline IL-13 level could be a potential predictive biomarker to stratify those who respond well to dupilumab. Furthermore, the simultaneous inhibition of IL-13 and IL-22 could be a promising alternative therapy for poor responders to dupilumab [69].

In conclusion, the previously unrecognized basophil-leukotriene axis, which is critical for acute itching flare-ups [19], could create the bases for new biologic drugs.

Thanks to these new sources of scientific evidence, we propose the use of a therapeutic flow-chart as a viable option as soon as other biologics, in addition to dupilumab, are approved in AD (Figure 2).

## 5. Chronic Urticaria

Acute urticaria is a very common condition in children, and typically self-heals in a few days or weeks [70,71]; a viral, allergic, food, or drug trigger can be often identified. On the contrary, chronic urticaria (CU) in children is less common. CU is defined by the daily presence of pruritic wheals, associated or not with angioedema, for over 6 weeks or with brief periods of well-being due to therapy. Wheals are well-circumscribed areas of non-pitting edema with blanched centers and raised borders that involve only the superficial portions of the dermis and are seen in conjunction with surrounding skin erythema [72]. Angioedema involves the submucosal surfaces of the upper respiratory and gastrointestinal tracts and deeper layers of the skin, including subcutaneous tissue [73]. 

In a child with urticaria onset, it is not possible to establish in which cases it will last over 6 weeks. To date, no predictive markers have been identified for the pediatric population [74]. A few studies performed in children suggest a lifetime prevalence of CU of 0.8% and an annual incidence of 0.6 to 2.1/1000, with no gender difference [75,76,77]. 

Based on the triggering factor, CU in children is classified as spontaneous (CSU) or inducible (CIU). In CSU, no external cause is found. In CIU, one or more triggers (often physical agents) can be identified through history and/or laboratory tests [71,78,79]. The terms “spontaneous” and “idiopathic” are often used as synonyms, although the definition of CSU is to be preferred as autoantibodies are often found in the serum of children [78]. However, several studies involving adults have not found any histological differences between CSU and autoimmune CU, although autoimmune CU can have a more severe and prolonged evolution; in addition, no evidence was found in children [80,81,82].

Concerning the natural history of CU in children, remission ranges from 10 to 32% 1 year after CSU onset and from 30 to 50% of cases 3 years after onset [83,84,85,86,87,88]. 

The pathophysiology of CSU is still not clear, although a disorder that causes the activation and degranulation of both mast cells and basophils, and the following release of preformed mediators (as histamine) and new formed mast cell products, has a pivotal role. Two major mechanisms were proposed to explain the pathogenesis of CU: the dysregulation of intracellular signaling pathways within mast cells and basophils and the development of autoantibodies against the high-affinity IgE receptor (FcεRIα) or IgE on both mast cells and basophils. The activation of FcεR1 is an important step in the development of urticaria and allergic disorders. This receptor is composed of an α-, a β-, and two γ- subunits [89]. While the α-subunit binds to the Cε3 constant region of the IgE molecule, the β- and γ- subunits contain cell immunoreceptor tyrosine-based activation motifs (ITAMs) which, when phosphorylated, promote the activation of spleen tyrosine kinase (SYK) and recruit secondary molecules with the subsequent activation of other intracellular pathways, including the phosphoinositide-3 kinase (PI3K) pathway. All these mechanisms cause degranulation of mast cells and increase the likelihood of pathologic mast cell activation when inappropriately upregulated. On the other hand, an autoimmune etiology is recognized in up to 45% of CSU cases; therefore, the presence of autoantibodies against IgE and FcεR1α is the main hypothesis to explain the altered activation of mast cells and basophils in patients with CSU. As a matter of fact, Grattan et al. found that 7 of the 12 subjects (of whom 6 were females) mounted a positive wheal-and-flare reaction to intradermal autologous serum injection, and fewer of these patients reported disease exacerbation with the application of pressure when compared to patients with a negative injection test [90]. These findings suggested that the patients with a positive result were less likely to have an inducible urticarial syndrome [91]. 

Several studies suggested that T lymphocytes have a key role in the pathogenesis of CSU because their interactions with the surfaces of mast cells seem to stimulate the release of inflammatory mediators, including TNF-α [92]. The TNF-α release is responsible for the upregulation of several mast cell genes, such as matrix metalloproteinase 9 (MMP9) and tissue inhibitor of metalloproteinase 1 (TIMP1). Moreover, MMP9 and TIMP-1 plasma levels are high in patients with CU, and correlate with disease severity [93]. 

The role of circulating IgG antibodies against IgE and FcεR1 in the pathogenesis of CSU is widely accepted in the literature. Approximately 40% of patients with CSU have circulating antibodies against IgE and FcεR1 [94], with a higher frequency in CSU patients with a positive intradermal autologous serum injection reaction [95]. Anti-FcεRI antibodies are found more frequently than anti-IgE antibodies. Autoantibodies against FcεRI on the surface of dermal mast cells and basophils cause chronic stimulation and degranulation of these cells with an IgE-independent mechanism [72]. On the contrary, autoantibodies IgG-anti IgE may bind to and crosslink receptor-bound IgE on the surface of mast cells and basophils, thus leading to activation and degranulation of these cells. Interestingly, autoantibodies against FcεR1α were found in the sera of patients with other autoimmune skin conditions and even in healthy subjects, though a pronounced histamine-releasing activity in individuals without CSU was not shown [96]. 

Several studies suggest that subjects with IgE autoantibody-mediated CSU have a faster improvement in response to biological therapy with omalizumab than those with IgG-mediated disease, due to the mechanism of omalizumab that affects IgE levels and FcεR1 status [97]. Further investigation is required to determine how the presence of unique autoantibodies can predict the disease course and comorbidities associated with various subtypes of CSU as well as overall responsiveness to therapy. 

The diagnosis of CU is based on history, and the occurrence and duration of wheals, typically itchy, migrating, and fading with finger pressure. The duration of a single lesion is usually less than 24 h with episodes lasting over 6 weeks. Angioedema is characterized by non-erythematous oedema, associated with a burning or pain sensation lasting up to 72 h, often located in the face, genitalia and extremities. There is no instrumental or laboratory test to diagnose CU [98].

Very often, currently available therapies for CSU do not achieve complete symptom control, further affecting quality of life [99]. Therefore, biological drugs represent an alternative treatment for those children with CSU. Nevertheless, guidelines concerning biologic drug use in children with CSU were generally extrapolated from adult studies [100] and RCT are still lacking in pediatric populations.

## 6. Biologics in Chronic Spontaneous Urticaria

The binding of IgE to FcεRI, which leads to intracellular signaling and the activation of mast cells, represents an important target of biological drugs [101]. In fact, the binding of free IgE to omalizumab prevents it from attaching to the FcεRI on mast cells and basophils [101]. Several trials demonstrated the efficacy and safety of omalizumab in CSU, confirmed in a recent systematic review [102]. Therefore, omalizumab is the only non-antihistamine drug approved for CSU therapy [103]. It is recommended for teenagers > 12 years who are unresponsive to antihistamine therapy, at 150 mg or 300 mg every 4 weeks [104].

Over time, an effort was made to search a predictive marker of response to treatment. Recently, a retrospective multicenter study of 470 adult CSU patients treated with omalizumab for 24 weeks showed that total serum IgE levels and their change could predict responses to treatment [105].

Ligelizumab is a new humanized monoclonal antibody against IgE. It seems to be promising, showing a better suppression of free IgE [106] and a better clinical improvement in CSU patients after 12 weeks of therapy compared to omalizumab [99]. Specifically, Maurer et al. [99] demonstrated that 72 mg of ligelizumab administered subcutaneously every 4 weeks resulted in complete clinical response in 51% of subjects, whereas 26% of the patients treated with 300 mg of omalizumab had the same response. Furthermore, side effects or laboratory abnormalities were represented by mild or moderate injection-site reactions and by mild injection-site erythema, and they were not dose-limiting. Despite these exciting findings, larger and longer trials are needed to establish the clinical efficacy and the safety profile of ligelizumab in patients with CSU.

Benralizumab is a monoclonal antibody that acts against the IL-5-receptor alpha. A recent trial enrolling 12 patients with CSU who were unresponsive to second-generation H1-antihistamines found a complete response (UAS7 = 0) in five patients and a partial response (UAS7 of six or lower) in two patients after 24 weeks of treatment with benralizumab. Furthermore, no drug-related side events were reported during the study, suggesting a good safety profile. These findings support the use of benralizumab in the treatment of CSU cases that are unresponsive to second-generation H1-antihistamines and provide evidence of a pathogenic role for infiltrating eosinophils [107]. 

Canakinumab is an IL-1β antagonist that is effective in cryopyrin-associated periodic syndromes associated with urticarial symptoms. A recent trial involving 20 patients with moderate-to-severe CSU with a 1:1 randomization to either canakinumab or placebo found no effect on lesions of CSU, suggesting the low contribution of IL-1β in the pathogenesis of CSU [108].

## 7. Conclusions Regarding the Use of Biologic Drugs for Chronic Urticaria in Children

New insights concerning the pathogenesis of—and the molecules involved in—chronic spontaneous urticaria have permitted the development of several biologic drugs designed to interfere with the underlying inflammatory pathway.

Although many biologic drugs are under investigation, omalizumab is currently the only monoclonal antibody approved in patients with severe and treatment-refractory CSU, and this is also the case in children. Its efficacy and safety were widely demonstrated, but the accessibility and the high cost represent a barrier to their use. Furthermore, the optimal duration of treatment is yet to be defined.

Ligelizumab showed higher affinity compared to omalizumab and, therefore, it seems to be a promising alternative, but its efficacy and safety have yet to be evaluated, especially in children.

It is desirable that, in the near future other biologics, will be approved, possibly with lower costs that could permit wider use. It is also desirable that the pediatric population is included in RCTs, despite the lower frequency of CSU in children compared to adults.

Table 1 shows the main characteristics of biologic drugs that are either currently used or under study in children with AD and chronic urticaria. 

## Figures and Tables

**Figure 1 biomedicines-09-01615-f001:**
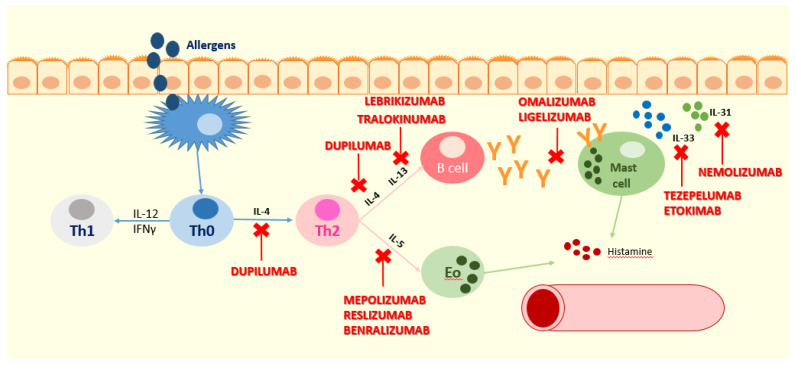
The point of action of the inhibitory activity of the mentioned biologic drugs. Dupilumab acts against the alpha subunit of the IL-4 receptor (IL4Rα), blocking IL-4 and IL-13 receptors and, therefore, the type-2 inflammation pathway. Mepolizumab is an anti-IL-5 antibody and acts by blocking Th2 cytokine expression mediated by IL-5 and eosinophils. Tezepelumab binds TSLP, preventing its interaction with the receptor complex, and etokimab binds IL-33; therefore, tezepelumab and etokimab act upstream of effector Th2 cytokines such as IL-4, IL-13, and IL-31. Nemolizumab is a humanized monoclonal antibody that acts against the IL-31 receptor, and it is essential in mediating itching. Lebrikizumab and tralokinumab are monoclonal antibodies that bind soluble IL-13, preventing heterodimerization of IL-13Ra1/IL-4Ra and the following signaling, are they are implicated in the Th2 immune response. Ligelizumab and omalizumab are monoclonal antibodies that act against IgE. Benralizumab is a monoclonal antibody that acts against the IL-5-receptor alpha.

**Figure 2 biomedicines-09-01615-f002:**
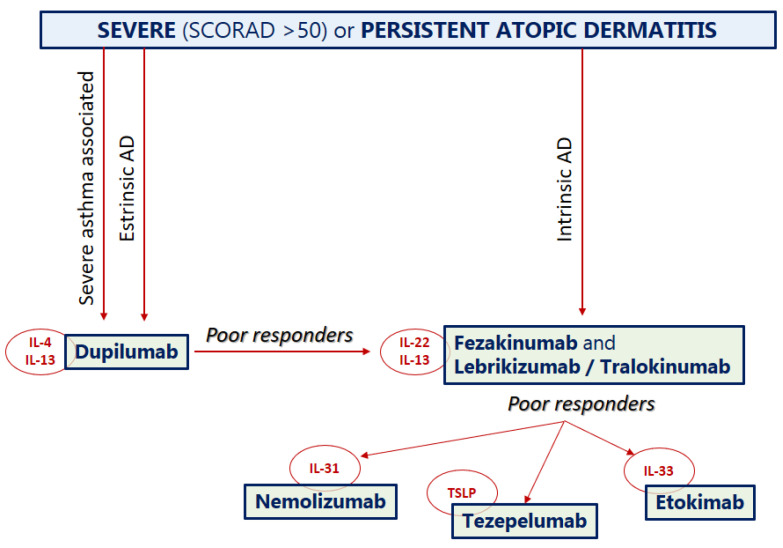
A proposal of a hypothetical therapeutic flow-chart for atopic dermatitis that could be applied as soon as other biologics, in addition to dupilumab, are approved.

**Table 1 biomedicines-09-01615-t001:** Main biologics currently approved and under study for allergic skin diseases in children.

Name	Mechanism Of Action	Age (Years)	Indications	Dosage	References
**Omalizumab**	Anti-IgE	≥12	CSU	300 mg SC every 4 weeks	[99,100]
**Dupilumab**	Anti-IL-4Rα	≥6	Moderate-to-severe AD	**Weight < 60 kg** 6–11 years old: 300 mg SC at day 1, at day 15 and successively every 4 weeks >12 years old: 400-mg loading dose SC followed by 200 mg every 2 weeks **Weight ≥ 60 kg:** 600 mg loading dose SC followed by 300 mg SC every 2 weeks	[18,27,38]
**Mepolizumab**	Anti-IL-5	/	AD: not approved	Phase I RCT	[43,44]
**Tezepelumab**	Anti-TSLP	/	AD: under study	Phase IIA RCT	[46]
**Etokimab**	Anti-IL-33	/	AD: under study	Phase I RCT	[47]
**Nemolizumab**	Anti-IL-31R	/	AD: under study	Phase IIB RCT	[51]
**Lebrikizumab**	Anti-IL-13	/	AD: under study	Phase IIB RCT	[53]
**Tralokinumab**	Anti-IL-13	/	AD: under study	Phase III RCTs	[55]
**GBR 830**	Anti-OX40	/	AD: under study	Phase II RCT	[56]
**Fezakinumab**	Anti-IL-22	/	AD: under study	Phase IIA RCTs	[57,58]
**Baricitinib**	Anti-JAK1 & 2	/	AD: under study	Phase III RCT	[60,61]
**Upadacitinib**	Anti-JAK1	/	AD: under study	Phase IIB RCT	[63,64]
**Abrocitinib**	Anti-JAK1	/	AD: under study	Phase IIB RCT	[65]
**Ligelizumab**	Anti-IgE	/	CSU: under study	Phase I RCT	[96,102]
**Benralizumab**	Anti-IL-5Rα	/	CSU: under study	Phase I RCT	[103]
**Canakinumab**	Anti-IL-1β	/	CAPS + CSU: under study	Phase II RCT	[104]

SC = subcutaneously; AD = atopic dermatitis; CAPS: cryopyrin-associated periodic syndromes; CSU = chronic spontaneous urticaria; RCT: randomized clinical trials.

## Data Availability

Not applicable.

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
