# Peer review of "Immunological Targets of Biologic Drugs in Allergic Skin Diseases in Children"

_biomedicines, 2021, doi:10.3390/biomedicines9111615_

Round 1

Reviewer 1 Report

The contents of this paper are related to atopy, but the contents are not specific and it is thought that more in-depth contents are needed. 

Author Response

We thank the reviewer for this relevant comment.

We have better defined the immunological mechanisms underlying the pathogenesis of atopic dermatitis to better explain the mechanism of action of the mentioned biological drugs. You can find the added sentences in lines 88-100, 118-132, 135-142, 293-298, 325-327.

Furthermore, we have improved the English language in the manuscript.

We have better defined the clinical implications in the “Conclusions” and we have proposed an hypothetic therapeutic flow-chart for atopic dermatitis as soon as other biologics besides dupilumab are approved in figure 2, after the conclusions.

Reviewer 2 Report

Dear Authors,

in my opinion the manuscript can be publish in this present form.

Author Response

We thank the reviewer for this relevant comment.

Reviewer 3 Report

The authors conduct a review article and aimed to investigate the effect of immunological targets of biologic drugs (16 monoclonal antibodies: 13 on study, 1 not approved for atopic dermatitis (AD), 1 for AD, 1 for chronic spontaneous urticaria (CSU)) in allergic skin diseases (atopic dermatitis and urticaria) in children.

Comments:

1.A brief description of the biologic drug mechanisms should be provided in Figure 1.

2.What are the clinical implications of this study?

3.Which kind of therapy would you suggest?

Author Response

We thank the reviewer for this relevant comment. We have tried to better define the immunological mechanisms underlying the pathogenesis of atopic dermatitis to better explain the mechanism of action of the examined biological drugs. You can find the added sentences in lines 88-100, 118-132, 135-142, 293-298, 325-327. Furthermore, we have provided a brief description of the biologic drugs mechanism in Figure 1, as suggested.

We have better defined the clinical implications in the “Conclusions” and we have proposed in figure 2 a hypothetic therapeutic flow-chart for atopic dermatitis as soon as other biologics besides dupilumab are approved.

Reviewer 4 Report

This article entitled “Immunological targets of biologic drugs in allergic skin diseases in children” is well reviewed on biologic therapies to treat atopic dermatitis and urticaria.

In addition to the molecular mechanisms, the cellular players of atopic dermatitis such as basophils but not mast cells should also be introduced (Ref.).

Ref.

A basophil-neuronal axis promotes itch.

Wang F, et al. Cell. 2021 Jan 21;184(2):422-440.

Basophils add fuel to the flame of eczema itch.

Mali SS, Bautista DM. Cell. 2021 Jan 21;184(2):294-296.

Author Response

We thank the reviewer for this relevant comment. As suggested, we have added the role of basophils in generating pruritus in atopic dermatitis. You can find the added sentences in line 72-84 and 130-132 and the added references (Mali SS, Bautista DM. Basophils add fuel to the flame of eczema itch. Cell. 2021, 21, 184, 294-296. And Wang, F; Trier, AM; Li, F; Kim, S; Chen, Z; Chai, JN et al.  A basophil-neuronal axis promotes itch. Cell. 2021, 184, 422-440.) are 18 and 19.

Furthemore, we have better defined the immunological mechanisms underlying the pathogenesis of atopic dermatitis to better explain the mechanism of action of the examined biological drugs. You can find the added sentences in lines 88-100, 118-132, 135-142, 293-298, 325-327.